# Uncertainty in Graph Neural Networks: A Survey

**Fangxin Wang, Yuqing Liu, Kay Liu, Yibo Wang, Sourav Medya, Philip S. Yu**
*{fwang51, yliu363, zliu234, ywang633, medya, psyu}@uic.edu*
*Department of Computer Science, University of Illinois Chicago*

**Reviewed on OpenReview:** *https://openreview.net/forum?id=0e1Kn76HM1*

## Abstract

Graph Neural Networks (GNNs) have been extensively used in various real-world applications. However, the predictive uncertainty of GNNs stemming from diverse sources such as inherent randomness in data and model training errors can lead to unstable and erroneous predictions. Therefore, identifying, quantifying, and utilizing uncertainty are essential to enhance the performance of the model for the downstream tasks as well as the reliability of the GNN predictions. This survey aims to provide a comprehensive overview of the GNNs from the perspective of uncertainty with an emphasis on its integration in graph learning. We compare and summarize existing graph uncertainty theory and methods, alongside the corresponding downstream tasks. Thereby, we bridge the gap between theory and practice, meanwhile connecting different GNN communities. Moreover, our work provides valuable insights into promising directions in this field.

## 1 Introduction

Graph Neural Networks (GNNs) have been widely employed in important real-world applications, including outlier detection (Liu et al., 2022b), molecular property prediction (Wollschläger et al., 2023), and traffic regularization (Zhuang et al., 2022). Within these domains, GNNs inevitably present uncertainty towards their predictions, leading to unstable and erroneous prediction results. Uncertainty in GNNs arises from multiple sources, for example, inherent randomness in data, error in GNN training, and unknown test data from other distributions. To mitigate the adverse impact of predictive uncertainty on GNN predictions, it is crucial to systematically identify, quantify, and utilize uncertainty. Uncertainty in GNNs plays a vital role in prevalent graph-based tasks, such as selecting nodes to label in graph active learning (Cai et al., 2017) and indicating the likelihood of a node being out-of-distribution (OOD) (Zhao et al., 2020). Furthermore, this practice enhances the predictive performance of GNNs (Vashishth et al., 2019), and bolsters their reliability for decision-making, such as making GNNs robust to attacks (Feng et al., 2021) and explains which components of the graphs influence GNN predictions (Ying et al., 2019).

Several challenges need to be addressed to boost GNNs with uncertainty. Firstly, different sources of uncertainties in GNNs need to be identified, as uncertainty may stem from multiple stages including data acquisition, GNN construction, and inference (Gawlikowski et al., 2023). Secondly, different categories of uncertainties should be matched with corresponding tasks. For instance, distributional uncertainty should be applied to graph OOD detection (Gui et al., 2022). Thirdly, uncertainties in GNNs are difficult to quantify correctly due to the lack of ground truth and unified evaluation metrics. Lastly, quantified uncertainty may be adapted and combined with the other components to better serve the task, e.g., graph active learning (Zhang et al., 2021a). Multiple studies have endeavored to solve these challenges in specific tasks. In this paper, we provide a holistic view of these tasks through the lens of uncertainty.

To the best of our knowledge, existing surveys or benchmarks either focus on a broader field, i.e., uncertainty in Deep Neural Networks (Abdar et al., 2021; Hüllermeier & Waegeman, 2021; Gawlikowski et al., 2023) and trustworthy GNN (Wu et al., 2022), or on narrower topics, e.g., uncertainty in spatio-temporal GNNs (Wu et al., 2021; Wang et al., 2023). Besides, these surveys mainly discuss how to estimate and measure uncertainty, missing concrete and practical guidance on how quantified uncertainties can be applied toward the final downstream task. Our survey aims to bridge these gaps by conducting a systematic review of GNNs from the perspective of uncertainty, with a special focus on incorporating uncertainty into graph learning. Furthermore, our survey contributes to the broader GNN community by

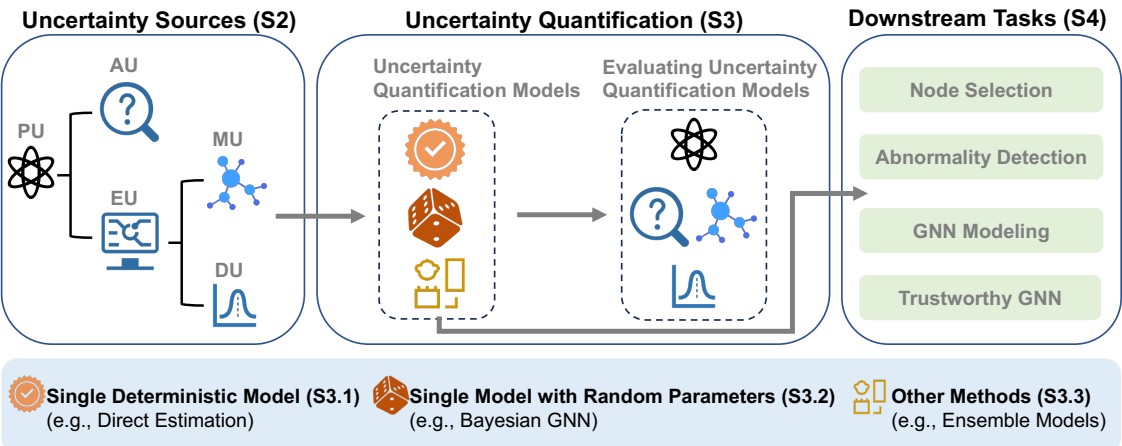

Figure 1: Overall Framework: *(1) identifying sources of uncertainty* (Section 2), *(2) quantifying uncertainty* (Section 3) and *(3) utilizing uncertainty for downstream tasks* (Section 4). In the first subfigure, 'PU' refers to predictive/total uncertainty. 'AU' represents aleatoric/data/statistical uncertainty, 'EU' is in short for epistemic uncertainty. 'MU' and 'DU' represent model and distributional uncertainty, respectively.

reviewing the trends of the current studies to approach uncertainty from distinct sub-communities, particularly across different downstream tasks. Our goal is to present a comprehensive framework that clarifies ambiguous terminologies, connects several communities, as well as offers valuable insights and directions to advance the overall development of GNN uncertainty and related fields.

In this paper, we organize the literature of uncertainty in GNNs into a three-stage framework, illustrated in Figure 1. Consequently, we introduce real-world applications utilizing GNN uncertainty in Section 5. Finally, in Section 6, we discuss significant and promising avenues for future research in this field. For further related background, we encourage readers to investigate Wu et al. (2020b) for a detailed introduction to GNNs, and Abdar et al. (2021); Hüllermeier & Waegeman (2021); Gawlikowski et al. (2023) for uncertainty in deep learning.

## 2    Identifying Sources of Uncertainty in GNNs

In machine learning, (total) uncertainty generally equals **predictive uncertainty (PU)**, defined as the uncertainty propagated into the prediction results. For accurate estimation and application of uncertainty, it is necessary to first identify different sources of uncertainty. Predictive uncertainty is commonly divided into aleatoric and epistemic ones based on their inherent different sources (Hüllermeier & Waegeman, 2021).

**Aleatoric uncertainty (AU).** Aleatoric uncertainty—known as data or statistical uncertainty—refers to variability in the outcome of an experiment which is due to inherent random effects. This often arises during the pre-processing stage, where data collection may lack sufficient information or contain noise and errors. As an example, collecting more data from the same graph distribution does not reduce aleatoric uncertainty; however, incorporating additional node features with non-overlapping information can reduce the inherent randomness. Unlike Munikoti et al. (2023), we align with the prevailing theory that aleatoric uncertainty can be estimated but is non-reducible through improvements in GNN (Hüllermeier & Waegeman, 2021), using it as the primary criterion to categorize existing literature.

**Epistemic uncertainty (EU).** In contrast, epistemic uncertainty—known as systemic uncertainty—refers to the uncertainty caused by the lack of knowledge of the GNN, thus reducible. It arises in the procedure of GNN modeling, further classified as **model uncertainty (MU)**. On one hand, model uncertainty comes from the choice of model structure. For instance, the shallow network structure may lead to the under-confidence of GNNs (Wang et al., 2022). On the other hand, the training process of GNN also introduces model uncertainty in the selection of parameters, for example, learning rate, epoch number, batch size, and techniques (e.g., weight initialization). Introduced by Malinin & Gales (2018), another type of epistemic uncertainty is **distributional uncertainty (DU)**, which arises when GNN has false assumption about the problem setting and data generation process. This is common for inductive inference, where GNN is trained on one graph and applied to another graph for prediction tasks, such that the training and test sets do

not come from the same distribution. While applying the trained GNN to the test data for inference, the distribution shift contributes to the uncertainty of prediction.

Note that since EU = MU + DU, and PU = AU + EU, then PU = AU + MU + DU. This relationship is illustrated in the middle subfigure of Figure 2. In most settings, prediction results are typically considered to have no distributional uncertainty, DU=0, thus EU=MU+0=MU. To avoid confusion of terms, we will use epistemic uncertainty (EU) at most time in Sections 4 and 5. If distributional uncertainty (DU) is explicitly stated, such as in tasks like OOD, we will use MU and DU.

Besides these frequently classified types, GNN uncertainty can also be classified from subjective viewpoints, e.g., vacuity (conflicting evidence) and dissonance (lack of evidence) (Zhao et al., 2020).

In general, the goal for GNN modeling should be to reduce EU, including delicate design of model structures and parameters, along with enhanced robustness and generalization to diverse distributions. On the contrary, since AU can only be quantified but not reduced, separating it from the epistemic uncertainty would provide a more reliable illustration of GNN performance.

# 3   Quantifying Uncertainty in GNNs

In this section, we classify existing GNN uncertainty quantification methods into three classes according to the number of models and whether model parameters are random variables. Following this classification, we present evaluation metrics for different quantification methods. In Table 1, we summarize and compare attributes of these methods.

Table 1: Comparisons of different families of GNN uncertainty quantification methods.

| Properties | Single Deterministic Model | | | Single Model with Parameters Being Random Variables | Ensemble Model |
|:---:|:---:|:---:|:---:|:---:|:---:|
| | Direct | Bayesian-based | Frequentist-based | | |
| **Number of Model(s)** | 1 | | | 1 | >1 |
| **Deterministic Parameters** | Yes | | | No, parameters with perturbation | No, different parameters |
| **Bayesian/Frequentist** | Neither | Bayesian | Frequentist | Bayesian | Either |
| **Parameter Sensitivity** | High | Lower | Lower | Low, affected by prior selection | Low |
| **Modeled DU** | No | Yes | No | No | No |
| **Separation of AU & EU** | No | Yes | No | Yes | Yes |
| **Modifications of Model** | No | Yes | No | Yes | Yes |
| **Cost of Training** | Low | | | Higher | High |
| **Cost of Inference** | Low | | | High | High |

## 3.1   Single Deterministic Model

This family of methods consists of a single model with deterministic parameters, i.e., each repetition of the method produces the same result. We classify them into direct, Bayesian-based, and Frequentist-based estimation methods.

**Direct estimation**. The most commonly used uncertainty estimation method is through the softmax probability of prediction outcomes, including maximum softmax probability (Wang et al., 2021) and entropy (Cai et al., 2017). Other heuristic measures, such as the distance of the sample to each class in the representation space (Xu et al., 2023), are also used to quantify uncertainty. Direct estimation methods integrate uncertainty estimation with prediction and do not require modifications of the original model.

However, it has been shown that GNN tends to be under-confident (Wang et al., 2021), which leads to inaccurate estimation of confidence and entropy. Confidence is an indication of how likely (probability) the predictions of a machine learning algorithm are correct, often computed as the maximum softmax class probability. Under-confidence, in other words, refers to that confidence, as the indication of accuracy, falls below the true prediction accuracy. To mitigate the issue of under-confidence, a well-calibrated GNN should produce predicted probabilities that accurately reflect the true likelihood of the predicted outcomes. This is usually evaluated with a smaller Expected Calibration Error (ECE) (Guo et al., 2017). To provide faithful uncertainty estimation, Wang et al. (2021) add a regularizer in

the GNN optimization objective, which penalizes the probabilities of the incorrect nodes and encourages the correct nodes to be more confident. Liu et al. (2022d) employ classical post-processing methods to calibrate GNN confidence, including histogram binning, isotonic regression, and Bayesian binning into quantiles. Similarly, Hsu et al. (2022b) propose a post hoc calibration method tailored for GNNs. Compared with uncalibrated direct estimation, the current calibration methods can effectively reduces ECE, but require additional calibration samples.

In addition, the inherent problem with estimated uncertainty is that the softmax output only reflects the total predictive uncertainty instead of the model uncertainty, leading to false estimation under the distribution shift. Therefore, direct estimation is a simple and easy-to-apply choice only when computation and data efficiency is preferable over estimation accuracy.

**Bayesian-based estimation** is a family of Bayesian-based deterministic models. Compared to direct estimation, the model fulfills the dual objectives of prediction and uncertainty estimation using Bayesian statistical principles. These methods typically utilize the Dirichlet distribution, which serves as the conjugate prior distribution of the categorical distribution in Bayesian inference. Dirichlet distribution is advantageous in directly reflecting higher distributional uncertainty with flatter (less variant) logits over all categories. Therefore, it is utilized in GNN uncertainty estimation approaches such as Evidential neural networks (Zhao et al., 2020; Stadler et al., 2021; Bazhenov et al., 2023). Zhao et al. (2020) first apply Dirichlet-based methods to GNNs, which incorporates information about the local neighborhood through the shortest path distances and the node labels. Stadler et al. (2021) better model the network effects through diffusing Dirichlet parameters in graph propagation. As the training loss are modified for both prediction accuracy and uncertainty quantification, especially for DU, these methods are more suitable for specific tasks, e.g., OOD detection, but may not produce the best performance in other tasks, e.g., node classification (Bazhenov et al., 2023). For the same reason, another disadvantage of existing methods is that they can not be directly applied on already trained GNNs post-hoc.

**Frequentist-based estimation.** This is a recently adverted family of methods to achieve trustworthy uncertainty estimation. Frequentist methods are typically model-agnostic and estimate uncertainty through post-processing. This implies that they do not alter the model or directly offer uncertainty estimation alongside prediction. In contrast to Bayesian-based approaches, they exhibit computational efficiency with fewer additional parameters, and their estimation is distribution-free and more robust compared to the direct methods.

Kang et al. (2022) estimate uncertainty through the length of confidence bands, constructed with jackknife (leave-one-out) sampling (Efron, 1982) on training data and influence function to estimate the change in GNN parameters. Though this idea is similar to MC dropout in Section 3.2, it leaves out samples and modifies model parameters in a Frequentist manner, not assuming a prior distribution of model components. Early works (Bååw, 2022; Clarkson, 2023) employing conformal prediction framework to GNNs provide guaranteed uncertainty estimation with the size of prediction sets under different settings, but involve less specific designs for GNNs. Recent works (Huang et al., 2023a; Zargarbashi et al., 2023) focus on optimizing for smaller set sizes (i.e., better efficiency) while covering the true labels in sets with guaranteed high probability. To achieve this goal, Zargarbashi et al. (2023) propagate node labels in the calibration set, and Huang et al. (2023a) directly use the inefficiency of the calibration set as the optimization objective.

However, existing Frequentist-based methods quantify total predictive uncertainty and do not explicitly distinguish between AU and EU, as opposed to the belief that they only quantify AU (Qian et al., 2023). It is also noteworthy that all these methods are built on the data exchangeability assumption. To satisfy this assumption, current methods split the data in fixed modes, mostly via random sampling. Lunde et al. (2023) provide some other general sampling methods in graphs where exchangeability assumption is not violated. However, in these cases, some specific graph properties are required. The validity of methods using non-random data splits, e.g., inductive settings, rely on nonexchangeable conformal prediction techniques, especially weighted conformal prediction (Barber et al., 2023). This area of research still has ample space for further exploration, particularly regarding how to optimally choose the weights. In practice, exchangeability assumption is believed to be satisfied if the coverage rate meets the pre-determined level in the experiments. Though this is not a sufficient condition to support the assumption, this practice is reasonable as guaranteed coverage rate is part of the method's goal and evaluation metric. Furthermore, in current graph conformal prediction methods, the guarantee of uncertainty quantification is marginal coverage on the total test data or conditional on some specific groups. Conditioning on each sample requires uniform unconformity scores (Einbinder et al., 2022), which are nearly impossible. This hinders the application of conformal methods on most downstream tasks in Section 4.

### 3.2 Single Model with Parameters Being Random Variables

Models with *parameters being random variables*, mostly Bayesian GNNs, are another widely used family of GNN uncertainty quantification methods. A vast majority of these methods rely on Monte Carlo (MC) dropout to sample weights for variational inference. Zhang et al. (2019b) first develop a Bayesian neural network framework for graphs depending on the graph topology, where the observed graph is viewed as a realization from a parametric family of random graphs. Hasanzadeh et al. (2020) point out that the graph dropout techniques (Rong et al., 2019) can be utilized at the test time for uncertainty estimation, and advance Zhang et al. (2019b) by considering node features and adaptive connection sampling. Cha et al. (2023) discover that under the conformal prediction framework, Bayesian GNN with a tempered posterior can improve the efficiency of predictions. Markov Chain Monte Carlo (MCMC) sampling methods are another family of Bayesian methods that represent uncertainty without a parametric model. However, these models are rarely used in graphs due to their expensive computational costs (Wu et al., 2021).

Bayesian GNNs can separate AU and EU through metrics introduced in Section 3.4. Nevertheless, existing research in this domain primarily concentrates on enhancing uncertainty awareness and minimizing predictive uncertainty. Despite their effectiveness in enhancing prediction accuracy and providing uncertainty estimates, Bayesian GNNs often encounter challenges in devising meaningful priors for graphs and efficiently estimating uncertainty through repeated sampling during inference. Additionally, these models typically do not account for the distributional uncertainty (DU).

### 3.3 Other Methods

Different from the Bayesian GNNs that perturb parameters around a single optimal model, ensemble models derive the final prediction based on the predictions from a set of independent models with different parameters or even with different model structures. Ensemble GNNs have been shown to provide more accurate prediction and reliable estimation than the single models (Bazhenov et al., 2022; Wang et al., 2023; Gawlikowski et al., 2023). Meanwhile, ensemble GNNs offer an intuitive means of expressing predictive uncertainty by assessing the diversity among the predictions of the ensemble members. AU and EU could also be differentiated similarly as in Bayesian GNN methods. Despite their wide applications, most GNN ensemble models follow the deep ensemble framework (Lakshminarayanan et al., 2017) without specific modifications for both graphs and uncertainty quantification purposes. Compared with the two families above, ensemble methods are less used in GNNs mainly due to their high computational cost.

Test-time data augmentation, another type of uncertainty quantification method, has been widely used in other fields, e.g., computer vision (Ayhan & Berens, 2022). It involves augmenting multiple samples for each test sample and applying the model to all samples to compute the distribution of predictions. Then, AU is estimated with the entropy of the model prediction distribution for a test sample and its augmentations, and EU can also be estimated in the same way as Bayesian GNN methods. However, in GNNs, test-time augmentation has only been utilized for other purposes. Jin et al. (2022) first propose to modify the node feature and the graph topology at the test time for better OOD generalization. Similarly, Ju et al. (2024) use test time augmentation to mitigate the bias of GNN towards achieving better performance on high-degree nodes. However, test-time augmentation has not yet been explicitly applied for graph uncertainty quantification and thus, is not included in our primary framework. Nevertheless, its potential adaptation for graph uncertainty quantification warrants exploration. Given its post hoc nature, it requires no additional data and shows better computational efficiency during training compared to ensemble and Bayesian GNNs.

### 3.4 Evaluating Quantification Models

We discuss the evaluation measures to assess the quality of GNN uncertainty quantification models. As in Figure 2, evaluations are not limited to a specific type of quantification model but to each targeted uncertainty source. This is because a model can quantify multiple sources of uncertainty, and models can only be evaluated regarding a specific uncertainty source. Given the challenge of acquiring ground truth uncertainty, various metrics have been proposed, and there is no consensus on a unified metric (Lakshminarayanan et al., 2017; Gawlikowski et al., 2023).

**Predictive (Total) uncertainty**. As discussed above (Liu et al., 2022a; Zhao et al., 2020), maximum softmax probability (referred to as confidence in this paper) and entropy are widely applied to quantify the predictive uncertainty in the classification problems. They are also utilized as uncertainty measures of predictive uncertainty Smith & Gal (2018). Notably, entropy is often used as a measure of aleatoric uncertainty when applied on labeled dataset, e.g., in decision tree. However, entropy in GNNs is calculated on the predicted probability of each sample. We believe it

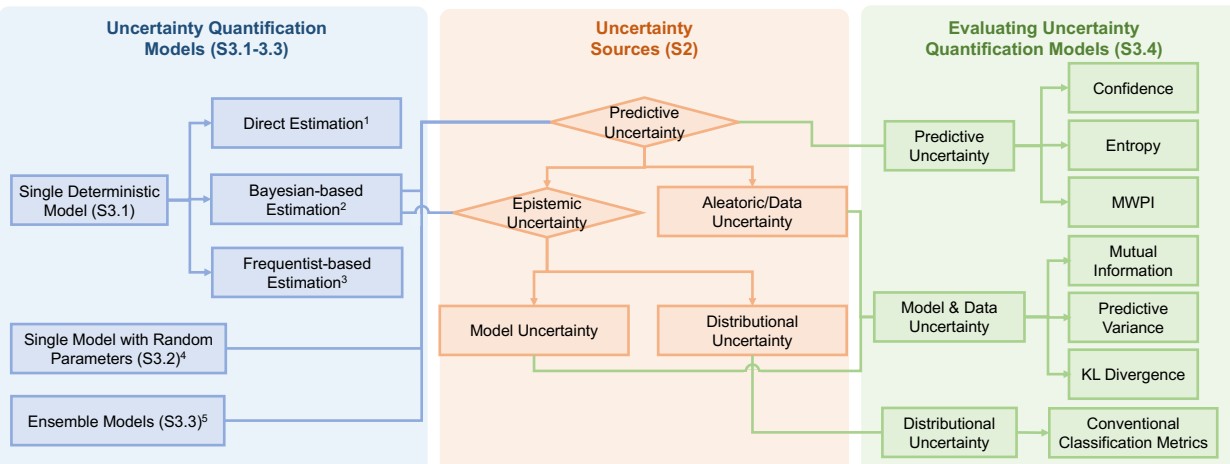

1. Wang et al. (2021); Cai et al. (2017); Guo et al. (2017); Hsu et al. (2022b); Xu et al. (2023)
2. Zhao et al. (2020); Stadler et al. (2021); Bazhenov et al. (2023)
3. Kang et al. (2022); Efron (1982); Huang et al. (2023a); Zargarbashi et al. (2023)
4. Zhang et al. (2019b); Hasanzadeh et al. (2020); Rong et al. (2019); Cha et al. (2023); Wu et al. (2021)
5. Bazhenov et al. (2022); Wang et al. (2023); Gawlikowski et al. (2023)

Figure 2: Bridge uncertainty quantification models and evaluation methods by uncertainty sources. The diamond shape represents the separation of uncertainty sources. Quantification models linked to any diamond indicate their ability to separate the corresponding uncertainty source. We merge "Model Uncertainty" and "Data Uncertainty" in evaluation as they are complementary and share similar evaluation metrics in some cases.

should be considered to represent predictive uncertainty, as opposed to aleatoric uncertainty evaluation metrics as in Gawlikowski et al. (2023). As mentioned in Section 3.1, these two metrics are vulnerable to distribution shifts and need calibration.

In regression problems, the mean width of prediction intervals (MWPI) (Kang et al., 2022; Huang et al., 2023a) can be used for evaluation. The coverage rate—the portion of ground-truth labels falling into these intervals—is verified to ensure the validity of this metric. Notably, efficiency is the same notion as MWPI in conformal regression, while in conformal classification, it refers to the average prediction set size under given coverage rate. In some works (Zargarbashi et al., 2023), this metric is accompanied by singleton hit rate, i.e., the prediction accuracy of prediction sets of size one. This joint use of both metrics satisfies the need for both prediction quality and uncertainty quantification. However, MWPI estimates the predictive uncertainty over a set of data but not on individual samples.

**Model & Aleatoric uncertainty**. The model uncertainty is commonly measured through mutual information (MI) to evaluate quantification models. MI is minimal when information in prediction does not increase with additional model knowledge. It is calculated as the difference between the entropy of the expected distribution (predictive uncertainty) and the expected entropy (aleatoric uncertainty) (Malinin & Gales, 2018; Zhao et al., 2020). Notably, MI is also referred to as information gain in some papers (Liu et al., 2022a). For Bayesian and ensemble GNNs, the predictive variance of sampled outputs is another option to interpret model uncertainty (Zhang et al., 2019a; Gal & Ghahramani, 2016). Furthermore, employed as one of the optimization objectives in some quantification methods (Liu et al., 2020; Zhao et al., 2020), the expected Kullback-Leibler (KL) divergence between approximated distribution and the posterior distribution can also evaluate model uncertainty. Given no distributional uncertainty, model uncertainty equals epistemic uncertainty. In this case, aleatoric uncertainty, namely data uncertainty, can be measured as the difference between the total uncertainty and model uncertainty; otherwise, the metric will be no longer meaningful.

**Distributional uncertainty**. It is believed that quantification of distributional uncertainty will be able to separate in-distribution (ID) and out-of-distribution (OOD) samples Malinin & Gales (2018). A popular approach to measure total distributional uncertainty is through classifying test samples. Samples with distributional uncertainty surpassing a given threshold are categorized as OOD samples, while those below the threshold are considered ID samples. In this, conventional classification metrics, e.g., Receiver Operating Characteristic (ROC) curve and Area Under Receiver

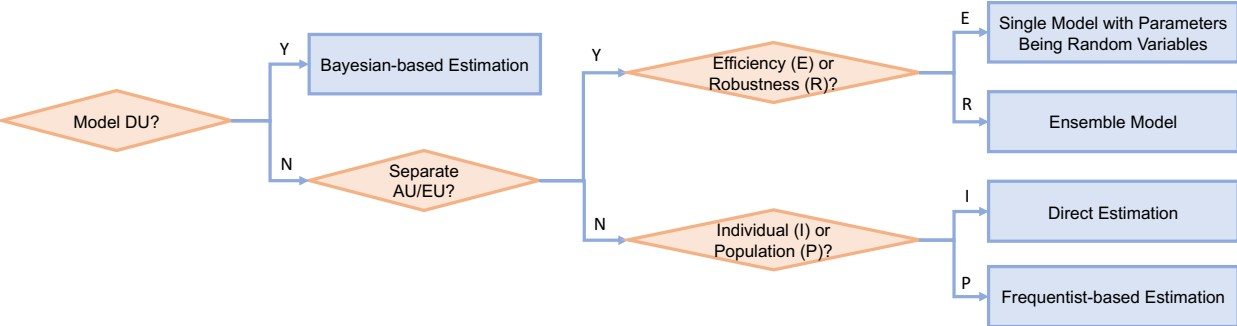

Figure 3: Flowchart illustrating recommended quantification methods for various conditions. Diamond shapes represent conditions, while rectangular shapes indicate the recommended methods.

Operating Curve (AUROC) (Zhao et al., 2020; Song & Wang, 2022; Gui et al., 2022), can be applied to measure the distributional uncertainty. However, this evaluation depends on the selection of test samples and threshold.

In conclusion, the effectiveness of existing metrics remains under exploration. As an alternative, many studies directly employ specific downstream tasks for evaluation to obtain reliable and straightforward measures. Nonetheless, depending on demands under different circumstances, we recommend diverse methods quantification methods according to their properties, as shown in Figure 3. First, it is crucial to identify the specific source of uncertainty being targeted. To our knowledge, only Bayesian-based estimation methods explicitly quantify distributional uncertainty. Subsequently, one must consider the necessity of separating aleatoric and epistemic uncertainty. If this separation is deemed necessary, two categories of methods are available: ensemble models, which consist of different member models, and single models with parameters treated as random variables. For computational efficiency, we recommend single models with parameters as random variables. However, for improved prediction performance and robustness across different distributions, ensemble models are preferable. In some cases, the separation of aleatoric and epistemic uncertainty does not yield significant benefits. For practitioners seeking highly computationally efficient methods for individual examples (e.g., nodes or links), direct estimation is adequate. Conversely, for a distribution-free and statistically guaranteed estimation of the overall test population, Frequentist-based estimation is recommended.

## 4 Utilizing Uncertainty in GNN Tasks

This section introduces four types of GNN tasks utilizing uncertainty, as Figure 4 describes. In addition to determining the type of uncertainty and how it is captured, we also focus on how uncertainty contributes to the task goal.

### 4.1 Uncertainty-based Node Selection

Node selection in graph learning refers to the process of selecting a subset of nodes from a graph that collectively contributes to specific tasks or analyses. Graph active learning and self-training are two common tasks involving node selection, where a collection of unlabeled data is labeled to improve the performance of the model (e.g., a GNN). Uncertainty is widely used as a criterion for node selection in these tasks.

**Active learning.** In graph active learning tasks, uncertainty and representativeness (diversity) are prevalent criteria for node selection (Ren et al., 2021). Uncertainty is often measured by the entropy of the prediction probabilities on the nodes. Cai et al. (2017) first propose a graph active learning framework, which uses a combination of entropy-measured uncertainty, information density, and node centrality to select nodes. Zhang et al. (2021a) estimate an ensembled uncertainty using the variation of prediction generated by committee models, which is more robust and reliable than entropy as a single deterministic model. The method selects top $k$ nodes to maximize the effective reception field using the topological information of the graph, excluding nodes that are less uncertain and less informative. Unlike the previous two methods, which select the most uncertain nodes independently, Zhang et al. (2021b) consider label propagation and select a subset of nodes with maximized uncertainty (entropy) reduction to include diversity. Labeled nodes can propagate their label information based on the graph distance and the node feature similarity, and then reduce the uncertainty of adjacent unlabeled nodes. Additionally, in a multi-task manner, Chang et al. (2024)

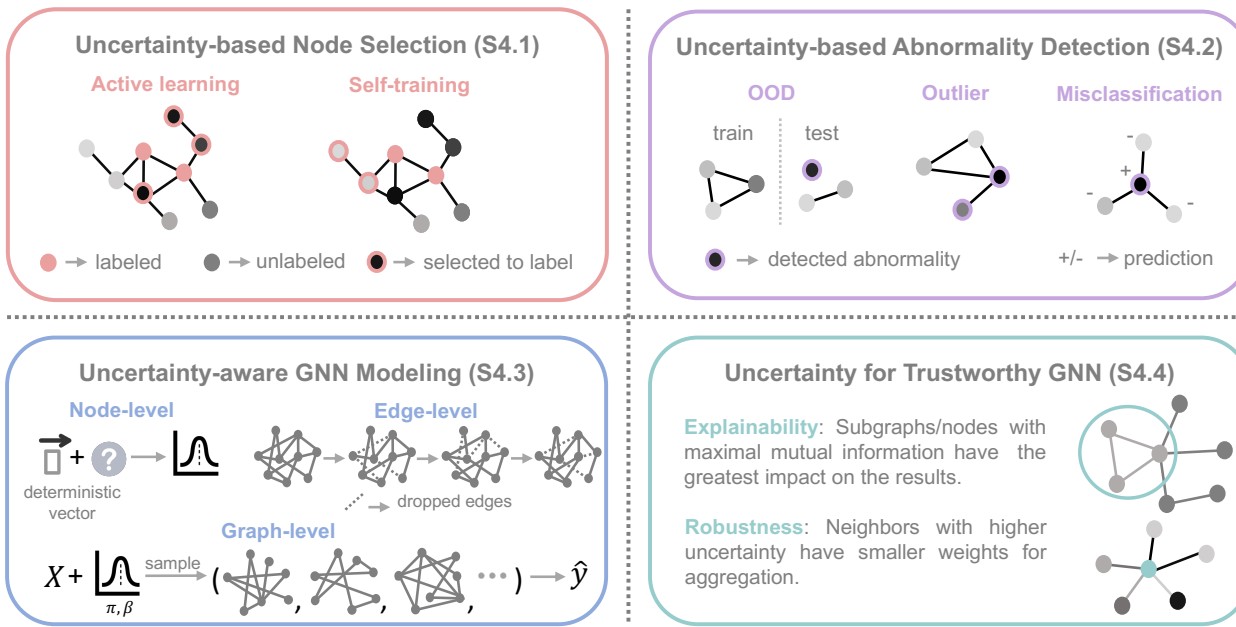

Figure 4: Illustration of representative usage of uncertainty in GNN tasks. The darker the color, the greater the uncertainty/weight.

estimate the confidence difference between the node classification task and anomaly detection task for node selection in graph active learning. Notably, in most cases, graph active learning methods that select nodes with only uncertainty can hardly outperform the ones with a combination of uncertainty and other measures.

To achieve dynamic and adaptive node labeling, reinforcement learning (RL) has been applied to graph active learning. Gao et al. (2018) uses the combination of the measures in Cai et al. (2017) to select nodes, but dynamically adjusts the combination weights based on a multi-armed bandit framework. Instead of measuring nodes independently, Hu et al. (2020) formulates the active learning problem as a sequential decision process on the graph considering the interconnections between nodes. Uncertainty is still measured with entropy but concatenated with other features, e.g., the divergence between a node's predicted label distribution and its neighbor's, to form the state representation of each node. The reward is the model performance on hold-out data updated with a trajectory of selected nodes. However, the RL-based active learning methods are significantly time-consuming to train.

**Self-training.** In graph self-training, i.e., pseudo-labeling, Li et al. (2018) propose one of the initial works that expand the labeled set with top $K$ confident nodes, quantified through maximal softmax probabilities. Considering that GNN is ineffective in propagating label information under the few-label setting, Sun et al. (2020) iteratively select and label the top $K$ confident nodes. Drawing upon the concept of ensembling to robustly select and label nodes, Yang et al. (2021) employs multiple diverse GNNs. Nodes are selected with confidence surpassing a pre-determined threshold and ensuring consistent prediction for labels across all models. Wang et al. (2021) is the first work to point out that the under-confidence of GNNs leads to biased uncertainty estimation, deteriorating the performance of uncertainty-based self-training node selection. With the calibrated confidence, they improve the performance of the self-training method introduced by Li et al. (2018). In consideration of graph structure information, Zhao et al. (2021) propose to compute entropy for each node, and add an entropy-aggregation layer to GNN for uncertainty estimation.

Similar to graph active learning, recent self-training methods (Liu et al., 2022a; Li et al., 2023b) consider combining uncertainty with representativeness, but from different perspectives. Liu et al. (2022a) believe the common self-training practice that only selects high-confidence nodes leads to the distribution shift from the original labeled node distribution to the expanded node distribution. Therefore, they follow the common practice that selects high-confidence nodes as pseudo-labels but decreases the weight of the nodes with low information gain in the training loss. Interestingly, the dropout technique employed to compute information gain is often interpreted as the model uncertainty, as introduced in Section 3.4. Similarly, Li et al. (2023b) also rely on the fact that the high-confidence nodes convey overlapping information, causing information redundancy in pseudo-labeled nodes. The representativeness of a node is

computed with the mutual information between the node and its neighborhood aggregation. Pseudo-labeled nodes are selected based on the representativeness and confidence scores exceeding predefined thresholds, with less confident nodes assigned lower weights in the gradient descent process. Further, Wang et al. (2024) reformulates the goal of selecting pseudo-labeled nodes as to reduce the uncertainty over all unlabeled nodes, and consider node selection in a combinatorial manner. This method can optimize the student model with a more global approach and reduces the correlation of selected nodes, improving representativeness and effectiveness of node selection in graph self-training.

Quantifying uncertainty accurately during the iteration process remains a significant challenge in both graph active learning and self-training tasks. One key issue is that the distribution of newly added pseudo-labeled data often diverges from the original training data distribution, making uncertainty estimates derived from the existing training data uncalibrated and noisy. In self-training, where teacher models frequently make inaccurate predictions, pseudo-labels can further introduce noise into uncertainty estimation. While limiting predictions to labeled data can reduce noise in some scenarios, it fails to capture the evolving uncertainty of updated models throughout the self-training iterations. Graph calibration models, as discussed in Section 3.1, offer a potential solution to this challenge by recalibrating uncertainty estimates. However, these models require a sufficient amount of labeled calibration data and typically involve longer training times. Moreover, in graph active learning, uncertainty serves primarily for node selection. By contrast, in self-training, it also represents the degree of noise in pseudo-labels. Unfortunately, inaccurate pseudo-labels can undermine the effective use of uncertainty, as even predictions with low uncertainty may contain incorrect labels, potentially leading to model misdirection during retraining.

In summary, uncertainty plays a crucial role in node selection tasks. Most of the existing methods rely on single deterministic models for estimating uncertainty directly, favoring computational efficiency in iterative training. For non-i.i.d. graph structures, node uncertainty is calculated alongside the influence of neighboring nodes and is often combined with other performance-enhancing metrics. Despite its widespread application, researchers should be cautious when estimating and applying uncertainty, particularly in graph self-training scenarios.

## 4.2 Uncertainty-based Abnormality Detection

Abnormality detection is a pivotal task in graph machine learning for safety and security. In this section, we define general abnormality detection as the task that identifies irregular patterns that deviate from the established norms. Uncertainty can serve as both a metric of abnormality and a tool for abnormality detection. Depending on the specific definition of abnormality, the application of uncertainty in abnormality detection can be systematically categorized into three distinct types: out-of-distribution detection, outlier detection, and misclassification detection.

**OOD detection.** To begin with, out-of-distribution (OOD) detection focuses on identifying instances that diverge from the training data distribution (in-distribution). OOD detection involves identifying samples from unknown classes and preventing models from making wrong decisions with overconfidence. In some literature (Wu et al., 2020a), OOD detection is integrated with the task of classification for in-distribution data and formulated as an open-world classification. Open-world classifiers recognize predefined classes seen during training and identify instances from new classes. For OOD detection, Wu et al. (2020a) maximize an entropy-based uncertainty loss for unlabeled nodes during training, such that the instances of unseen classes can have a low and balanced predictive probability over the already seen classes. During inference, if the maximal class probability of an instance is lower than a predefined threshold, the instance is regarded as OOD. In addition, Zhao et al. (2020) propose to quantify various types of uncertainty with graph-based Dirichlet distribution, and point out that vacuity uncertainty, derived from a lack of evidence or knowledge, is the most effective in OOD detection among other uncertainties. Furthermore, considering the non-i.i.d. nature of graph data, Stadler et al. (2021) perform Bayesian posterior updates for predictions on interdependent nodes through diffusing Dirichlet parameters. Epistemic uncertainty without network effects is shown effective in detecting random OOD nodes with perturbed features, while epistemic uncertainty with network effects performs better in identifying nodes from the left-out class not present in the training data.

**Outlier detection.** Apart from OOD detection, outlier detection (also known as anomaly detection) is another prevailing task that effectively leverages uncertainty in graph learning. Outlier detection refers to the identification of data that exhibit significantly different behaviors or characteristics compared to the majority of the graph. Different from the OOD data, outliers can also exist in the training data. Ray et al. (2021) directly estimate the predictive uncertainty with deviation from the distribution for interpretable anomalous behavior detection on multivariate time series. Notwithstanding conventional uncertainty quantification methods, Liu et al. (2022c) interpret the graph anomaly scores using

a Bayesian approach (Perini et al., 2020), which converts the anomaly score into the confidence level of the GNN in its outlier prediction for each example. The key idea of this conversion is to align the likelihood of an example being an outlier with its position in anomaly score distribution.

**Misclassification detection.** Moreover, uncertainty can also be utilized to identify instances with incorrect model prediction, known as misclassification detection. Zhao et al. (2020) attempt various types of uncertainty, and concludes that dissonance (i.e., uncertainty due to conflicting evidence) is the most effective in misclassification detection. Besides detecting OOD nodes, Stadler et al. (2021) also utilize their proposed method to detect misclassified nodes. In most experiments, aleatoric uncertainty achieved the best performance, rather than epistemic uncertainty in OOD detection. This observation supports the necessity to separate and utilize task-specific sources of uncertainty.

In summary, recent studies emphasize the importance of uncertainty in graph-based anomaly detection, although there is inconsistency in the type of uncertainty utilized for the same purpose. For instance, both predictive uncertainty (Wu et al., 2020a), epistemic uncertainty (Stadler et al., 2021), and vacuity uncertainty (Zhao et al., 2020) have been applied to detect OOD nodes. There is a lack of a systematic framework for selecting quantification methods and applying estimated uncertainty in abnormality detection. Establishing such a framework would be advantageous for both the theoretical exploration and the practical application of abnormality detection.

### 4.3 Uncertainty-aware GNN Modeling

Graphs serve as powerful representations of complex relationships and structures, but their structural uncertainty poses challenges for effective learning and inference tasks. Fortunately, GNN can improve its prediction performance through uncertainty-awareness. Based on addressing uncertainty from different components of graph structure, related works can be divided into three categories: (1) node-level, (2) edge-level, and (3) graph-level.

**Node-level.** In graphs, nodes often exhibit conflicting or contradictory characteristics, such as belonging to different communities or revealing contradictory underlying patterns (Bojchevski & Günnemann, 2017). This discrepancy should be reflected in the uncertainty of node embeddings, highlighting the need to account for uncertainty at the node-level in GNN structures. Hajiramezanali et al. (2019) incorporate stochastic latent variables by combining a Graph Recurrent Neural Network (GRNN) with a Variational Graph Autoencoder (VGAE) (Kipf & Welling, 2016). The proposed variational GRNN is designed to capture the temporal dependencies in dynamic graphs. Moreover, it aims to represent each node by modeling its distribution in the latent space rather than assigning it a deterministic vector in a low-dimensional space, thus enhancing its ability to represent the uncertainty associated with node latent representations. Vashishth et al. (2019) propose ConfGCN, which introduces confidence (inversely proportional to uncertainty) estimation for the label scores on nodes to enhance label predictions. More specifically, ConfGCN assumes a fixed uncertainty for input labels and defines an influence score of each node as a parameter that can be optimized jointly with the network parameters through back-propagation. The influence score here is the inverse of the co-variance matrix-based symmetric Mahalanobis distance between two nodes in the graph. Sun et al. (2021) generate stochastic representations using a hyperbolic VGAE, combining temporal GNN to model graph dynamics and its uncertainty in hyperbolic space. Xu et al. (2022) introduce a Bayesian uncertainty propagation method and embeds GNNs in a Bayesian modeling framework. This approach uses Bayesian confidence of predictive probability based on message uncertainty to quantify predictive uncertainty in node classification. The proposed uncertainty-oriented loss penalizes the predictions with high uncertainty during the learning process, enabling GNNs to improve prediction reliability and OOD detection ability. Liu et al. (2022e) introduce the concept of a mixture of homophily and heterophily at the node-level, revealing that GNNs may exhibit relatively high epistemic uncertainty for heterophilous nodes. To address this issue, the paper proposes an Uncertainty-aware Debiasing framework, which estimates uncertainty in Bayesian GNN output to identify heterophilous nodes and then trains a debiased GNN by pruning biased parameters and retraining the pruned parameters on nodes with high uncertainty.

**Edge-level.** To better consider the inter-dependencies in the graph structure, some methods also explore uncertainty beyond the node-level. While Veličković et al. (2018) has applied a similar dropout technique on edge attentions, Rong et al. (2019) first formally presented the formulation of DropEdge. It randomly removes a specified number of edges from the graph by drawing independent Bernoulli random variables (with a constant rate) during each training iteration. Rong et al. (2019) theoretically and empirically show that DropEdge, as a data augmenter and a message-passing reducer, can alleviates the problems of over-fitting and over-smoothing. Hsu et al. (2022a) propose two edgewise metrics for graph uncertainty estimation: (1) edgewise expected calibration error (ECE) is designed to estimate the

general quality of the predicted edge marginals, utilizing graph structure and considering confidence calibration of edgewise marginals; (2) agree/disagree ECEs extend the idea of edgewise ECE by distinguishing homophilous and heterophilous edges.

**Graph-level.** Some works address uncertainty in the overall structure or properties of the entire graph. Zhang et al. (2019b) is the first work to develop the Bayesian neural network framework tailored for graphs, treating the observed graph as a realization from a parametric family of random graphs. The joint posterior distribution of the random graph parameters and the node labels are then targeted for inference. However, besides the considerable computation cost, Bayesian GNN relies on the selection of the random graph model, which may vary across different problems and datasets. Hasanzadeh et al. (2020) introduce Graph DropConnect (GDC), a general stochastic regularization technique for GNNs through adaptive connection sampling, which yields a mathematically equivalent approximation of Bayesian GNNs training. This unified framework mitigates over-fitting and over-smoothing issues of deep GNNs, while also prompting uncertainty-aware learning. While both Bayesian GNN and GDC are aware of graph-level uncertainty, the current implementation of posterior inference in Bayesian GNN solely relies on the graph topology and neglects the node features. In contrast, GDC involves a free parameter that allows adjustment of the binary mask for edges, nodes, and channels, thereby facilitating flexible connection sampling and addressing uncertainty from various levels of graph structure. Therefore, GDC can be viewed as an extension and generalization of DropEdge, which only focuses on edge-level. Moreover, unlike stochastic regularization methods relying on fixed sampling rates or manual hyperparameter tuning (e.g., DropEdge), GDC enables joint training of drop rate and GNN parameters. Liu et al. (2020) propose an Uncertainty-aware Graph Gaussian Process (UaGGP) approach to address uncertainty issues in graph structures, especially when edges connect nodes with different labels, leading to predictive uncertainty. The work employs graph structural information for the feature aggregation and the Laplacian graph regularization. Leveraging the randomness modeled in the Gaussian Process, UaGGP introduces graph Laplacian regularization based on Gaussian Process inferences and symmetric Mahalanobis distance to tackle structural uncertainty.

To address the uncertainty in graph structure, many studies leverage Bayesian inference to capture uncertainty and estimate model parameters. Some approaches can handle over-smoothing and over-fitting phenomena in GNN simultaneously. In addition, we find these often hold the assumption that edges connecting nodes with different labels would result in misleading neighbors and causing uncertainty in the predictions. Although this assumption is shown valid and effective, we believe it is not the only factor causing uncertainty in graph structure, and this hypothesis is also not suitable for heterogeneous graphs. It would be interesting to design methods that treat structural uncertainty from other geometric information from the graph, e.g., nodes connecting to different communities (Huang et al., 2023b). Furthermore, while existing works primarily aim to develop uncertainty-aware GNNs to enhance prediction performance, a few explicitly (1) evaluate the reduction of uncertainty in GNNs compared to prior network structures, and (2) validate as well as explain how the reduction of uncertainty contributes to improving prediction accuracy. Finally, the methods in this domain rarely compare against each other as baselines, lacking a universal benchmark.

## 4.4 Uncertainty for Trustworthy GNN

Trustworthy GNNs aims to provide explainability of decisions made by GNNs and to be robust to noise and adversarial attacks to ensure the integrity and security of the model. Uncertainty can serve as a tool to build algorithms for the interpretation of GNNs decisions. Additionally, considering uncertainty during the aggregation mechanisms in GNNs can enhance robustness. Thus, uncertainty quantification can be an effective approach to achieving trustworthy GNNs.

**Explainability.** Uncertainty quantification provides insights for the reliability and confidence of GNN decisions. Specifically, mutual information explains the relationship between the GNNs predictions and the components that impact GNNs predictions. Ying et al. (2019) propose GNNEXPLAINER which identifies a subgraph and/or node features that maximize mutual information with the prediction, to explain which components of the graph have the most impact on the prediction results. Mutual information is nonparametric and does not require prior knowledge and assumptions on models. Thus, it is model-agnostic and suitable for providing interpretable explanations for any GNN-based models. Besides, uncertainty quantification can also be used to explain predictions of automated GNNs. Yang et al. (2022) propose HyperU-GCN, which combines hyperparameter optimization with uncertainty quantification, to explain the selection of hyperparameters. It quantifies hyperparameter uncertainty as the mutual information between the model prediction and hyperparameters using a probabilistic hypernetwork. HyperU-GCN employs a concept akin to Bayesian GNNs for computing this mutual information, which marginalizes hyperparameters rather than model parameters. Unlike GNNEXPLAINER, HyperU-GCN requires model hyperparameter priors (e.g. normal distribution),

yet leveraging Bayesian GNNs enables it to better calibrate model predictions. There are many other methods to interpret GNNs besides using uncertainty, and we refer readers to Kakkad et al. (2023) for a comprehensive overview. For example, GraphLIME (Huang et al., 2022) samples neighbors and finds the most representative features as GNN local explanations based on Hilbert-Schmidt Independence Criterion Lasso; GCExplainer (Magister et al., 2021) applies an Automated Concept-based Explanation approach (Ghorbani et al., 2019), which maps node representations to the concept space, to extract important concepts as global GNN explanations.

**Robustness.** GNNs are vulnerable to adversarial attacks, i.e., small perturbations in graph structures and node attributes could lead to performance degradation. Incorporating uncertainty in graph learning empowers GNNs with robustness against such attacks. Zhu et al. (2019) propose Robust GCN (RGCN), which adopts Gaussian distributions as the hidden representations of nodes instead of certain vectors. By introducing Gaussian distributions, RGCN can automatically absorb the effects of abnormal adversarial changes and enhance its robustness. Besides, RGCN utilizes a variance-based attention mechanism, which assigns different weights to node neighborhoods according to their variances to reduce the propagated adversarial effects from neighbors. Though RGCN can model epistemic uncertainty in GNNs by Gaussian distribution variances, it does not consider aleatoric uncertainty and uncertainty from graph topology. Thus, Feng et al. (2021) propose quantifying uncertainties from graph topology features, measuring both epistemic and aleatoric (i.e., data) uncertainties. This paper utilizes the Bayesian GNN to estimate epistemic uncertainty and node diversity, which is intrinsic to the graph topology and defined as the number of different labels in the node's k-hop neighbors, to estimate aleatoric uncertainty. Another disadvantage of RGCN is that it relies on prior distributions and the uncertainties are not well-calibrated. Therefore, Shanthamallu et al. (2021) propose UM-GNN, which utilizes Bayesian GNNs with Monte Carlo dropout to estimate epistemic uncertainty of a standard GNN and assigns low uncertainty nodes with high attention. UM-GNN also utilizes an uncertainty-matching strategy to align prediction results between the standard GNN and a structure-free surrogate model to avoid simple prior assumptions on structural uncertainties. Besides, UM-GNN uses a label smoothing regularization to calibrate uncertainties. Zhang et al. (2021c) propose a multi-view confidence-calibrated GNN encoder (MCCG) module, which utilizes Dirichlet distribution to calibrate the prediction confidence for noisy or adversarial samples and utilizes the estimated uncertainty scores to control information propagation. Although these defense methods apply different quantification methods for different sources of uncertainty, they use the same key idea that increasing the weights of low-uncertainty nodes reduces the impact of the adversarial attack. Besides, although it is not explicitly stated that uncertainty quantification methods were used, some works have employed uncertainty-related approaches to verify the certifiable robustness of GNNs. For example, Bojchevski & Günnemann (2019) uses the minimum worst-case margin between the predicted or ground-truth node label and other labels under perturbations of graph structures to verify whether the node is robust. By intercepting messages from perturbed nodes, Scholten et al. (2022) uses whether the perturbed prediction is consistent with its neighbors regarding the majority vote to certify the robustness of GNNs. Both works perturb the original input graphs and measure the robustness with the variability of predictions under perturbations. This idea is similar to Bayesian GNNs, where uncertainty is measured with the variance of prediction with randomized GNN parameters. In this way, robust GNNs are expected to preserve consistent and certain predictions with noisy data. Sun et al. (2022) provides more GNNs robustness studies.

In conclusion, uncertainty can enhance the trustworthiness of GNNs from both explainability and robustness, with a consensus on how to utilize uncertainty. Other interesting directions are also worth exploring. For example, instead of increasing the weight of certain nodes, the robustness of GNN may also be improved by training with perturbations obtained by maximizing the model's estimated uncertainty (Pagliardini et al., 2022).

## 5 Uncertainty in GNNs: Real-world Applications

**Traffic.** GNNs have been widely used to model network traffic, especially traffic demand forecasting. Uncertainty arises in traffic systems both spatially and temporally, e.g., travel demand may increase on game days around the stadiums. Wu et al. (2021) compare six different Bayesian and Frequentist-based uncertainty quantification methods for road network traffic prediction problems, regarding computational efficiency, asymptotic consistency, etc. Zhuang et al. (2022) specifically focus on sparse travel demand and proposes STZINB-GNN, where a spatial-temporal embedding is designed with an additional sparsity parameter to learn the likelihood of inputs being zero. The probability layer in the embedding allows for prediction with a level of confidence and thus, quantifying the demand uncertainty. Wang et al. (2023) propos a framework of probabilistic GNNs, Prob-GNN, to quantify spatiotemporal uncertainty of traffic demand. Besides different GNNs, different probabilistic distribution assumptions with different properties

are compared, such as Laplace and Gaussian distribution, as well as single and ensembled Gaussian models. Prob-GNN can not only quantify spatiotemporal uncertainty but also generalize well under significant system disruption and identify abnormally large uncertainty. The experiment results show that the GNN performance is more impacted by probabilistic distribution assumptions than the selection of GNNs, underscoring the importance of incorporating randomness and uncertainty into the traffic prediction model. To quantify both aleatoric and epistemic spatiotemporal uncertainty, Qian et al. (2023) propose DeepSTUQ, which uses Bayesian dropout for pre-training and ensembling for re-training, and then calibrates the trained model with temperature scaling. In traffic applications, models are typically evaluated with point estimate error along with the mean prediction interval width (MPIW).

**Molecules.** Drug discovery is a graph-level problem of identifying and developing new molecules with desired properties, where each molecule is modeled as a graph. Zhang et al. (2019a) compare different Bayesian-based methods for accurate and reliable molecule property predictions in drug discovery. They reveal a few key aspects: (1) both epistemic and aleatoric uncertainty are correlated with error and improve the accuracy of selective prediction, thus, total uncertainty should be utilized for uncertainty estimation; (2) Bayesian uncertainty estimation could overcome the bias of the synthetic dataset by informing the user about the level of uncertainty; (3) the estimated uncertainty is also applied to guide active learning for drug discovery, showing better performance with initially diverse labeled classes. Similarly, Ryu et al. (2019) apply Bayesian GNN with Monte Carlo dropout to estimate uncertainty in activity and toxicity classification problems. Additionally, they use aleatoric uncertainty to evaluate the quality of the existing and synthetic datasets. To reduce overconfident mispredictions for drug discovery under distributional shift, Han et al. (2021) introduce CardioTox, a novel real-world OOD data benchmark with molecules annotated by Tanimoto graph distance and proposes a distance-awareness classifier, GNN-GP. Follow-up studies (Aouichaoui et al., 2022; Kwon et al., 2022) also derive similar conclusions. In molecular force fields, GNNs have also shown promising performance for quantum mechanical calculations to accelerate molecular dynamics trajectory simulations. Besides common concerns about prediction accuracy, uncertainty awareness, and computational costs, Wollschläger et al. (2023) propose physics-informed desiderata, including symmetry, energy conservation, and locality. In addition to evaluating previous uncertainty quantification methods, they propose Localized SVGP-DKL, which combines the Bayesian GNN with the localized neural kernel to satisfy the physics-related criteria.

**Others.** To segment images of complex geometry or topology, Gupta et al. (2023) employ GNNs to jointly reason about the structures (nodes) and capture the high-order spatial interactions (edges). A probabilistic model provides structure-wise uncertainty estimates, which could identify highly uncertain structures for human verification. In computational social systems, Zha et al. (2023) apply an uncertainty-aware graph encoder (UnGE) to represent job titles as Gaussian embeddings and capture uncertainties in real-world career mobility. In fault diagnosis, a Bayesian hierarchical GNN is proposed by Chen et al. (2023), where the total uncertainty is measured by the dropout. Using uncertainty as a weight, the loss encourages the feature of a highly uncertain sample to be temporally consistent for robust feature learning. Uncertainty in GNN is also widely applied in diverse domains, including energy (Li et al., 2023a), robotics (Gao et al., 2023), and recommendation systems (Zhao et al., 2023).

# 6 Future directions

**Identifying fine-grained graph uncertainty.** While current research on graph data has asserted their ability to differentiate between various types and sources of uncertainty (Section 2), only a few efforts have been made to identify and quantify uncertainty tailored specifically for graph components. For example, (Bojchevski & Günnemann, 2017; Liu et al., 2022e) have assumed that heterophily, the conflict between two connected nodes, is correlated with uncertainty in node classification. These attempts demonstrate that fine-grained graph uncertainty is essential for downstream tasks. Therefore, it is promising to develop other unexplored fine-grained uncertainties that contribute to such specific tasks. For instance, researchers might seek to identify specific types of graph distribution shifts, such as shifts in graph node features, labels, and graph structure. Decomposing distributional uncertainty at a more granular level could aid in distinguishing and leveraging the corresponding shifts, thereby enhancing OOD detection and generalization. Furthermore, different forms of finer-grained graph uncertainty should be comparable in scale. By comparing their corresponding uncertainty measures, we can determine whether the distributional uncertainty arising from node feature or graph topology, or the GNN modeling is dominant. This contributes to uncertainty estimation with improved explainability by indicating how each component in graph data, GNN modeling, and inference process contributes to the final GNN predictive uncertainty.

**Constructing ground-truth datasets and unified evaluation metrics.** The critical unresolved issue of the lack of ground-truth datasets and metrics for evaluation, as discussed in Section 3.4, persists. While aleatoric and distributional uncertainty can be artificially introduced into synthetic graph data with predefined underlying distributions or generating processes, synthetic data has difficulty in simulating complex real-world scenarios, with diverse and disentangled sources of uncertainty. Besides, it is also challenging to obtain ground-truth values for model uncertainty, as it depends on the selection of models. On the other hand, though humans could be involved in evaluating ground-truth aleatoric uncertainty for real-world data, biases may still arise due to varying levels of human cognitive abilities. Additionally, the lack of unified evaluation metrics for different tasks, such as classification or regression tasks, further complicates the evaluation process. Thus, uncertainty across different tasks is not directly comparable.

Currently, a feasible alternative is to assess whether uncertainty estimation enhances well-defined metrics for the final task performance rather than directly evaluating the quality of uncertainty estimation. However, there is a gap in understanding whether performance enhancements are solely due to the selection of quantification methods or if they ideally result from an appropriate match between quantification methods and corresponding uncertainty types, where the uncertainty type itself aids the task. Bridging this gap still requires further efforts.

**Quantifying task-oriented uncertainty.** The existing literature lack a systematic investigation aimed at identifying the most effective uncertainty method for downstream tasks introduced in Section 4. This is partly attributable to the observation that researchers in a specific domain tend to follow a specific method to quantify uncertainty. For instance, in graph active learning, entropy is frequently used for uncertainty estimation without additional elucidation. It remains unclear whether alternative uncertainty measures, such as mutual information, either independently or in combination with other selection criteria, would yield superior performance. Another interesting direction is whether the separation of aleatoric and epistemic uncertainty is necessary. For instance, in other active learning tasks, epistemic uncertainty measures generally outperform total uncertainty ones, while aleatoric uncertainty generally excels in failure detection (Kahl et al., 2024). However, in graph active learning and outlier detection, the common measures capture predictive uncertainty instead of separating it. Therefore, there is a need for systematic validation of various uncertainty types, quantification methods, and evaluation techniques across a wide range of pertinent downstream tasks.

**Designing better methods for real-world applications.** In Section 3, we have introduced multiple categories of uncertainty quantification methods and showed their strengths and weaknesses. In real-world applications, several aspects need to be taken into primary consideration. First of all, whether methods can effectively capture specific sources and types of uncertainty that are relevant to particular types of tasks. For example, Dirichlet-based methods directly model distributional uncertainty, thus showing superior performance over other methods in graph OOD detection. Secondly, computational efficiency and difficulty of application to existing prediction frameworks are also of vital importance. For instance, though ensemble models outperform most other methods in both model prediction and uncertainty estimation, they are rarely applied to real-world problems due to their high computational costs. One potential direction is to explore graph ensemble distillation (Zhang et al., 2020) as a cost-efficient alternative, which uses smaller GNNs trained on sub-problems and teaches one student GNN to handle OOD and estimate uncertainty (Hinton et al., 2015). Lastly, one should explore whether the proposed methods can consistently achieve reliable performance across diverse settings. For example, the maximal softmax probability is widely applied in real-world problems due to time efficiency but suffers from under-confidence and is subject to the effect of distributional shifts. In conclusion, it is an exciting direction to develop more efficient, easy-to-apply, robust graph uncertainty quantification methods that can identify diverse fine-grained sources and types of uncertainty.

# 7   Conclusions

In this paper, we have categorized the existing methods into a novel schema and highlighted the importance of identifying, quantifying, and utilizing uncertainty in GNNs for various downstream tasks. Specifically, we have emphasized the need to identify fine-grained uncertainty, especially graph-related uncertainty, and construct ground-truth datasets and unified evaluation metrics. Being the first survey to systematically review GNN uncertainty, we call for the development of more efficient, easy-to-apply, and robust uncertainty quantification methods, along with a systematic investigation and benchmark into existing methods on popular downstream tasks.

## Acknowledgments

This work is supported in part by NSF under grants III-2106758, and POSE-2346158.

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
