# OpenReview forum: "Uncertainty in Graph Neural Networks: A Survey"
_TMLR — Accepted by TMLR_

### Review · Reviewer_hKxZ · 2024-03-25

**Summary Of Contributions:**

The paper surveys the source, measure and modeling regarding uncertainly in GNN. It summarizes uncertainly of GNNs in various aspects, categorizing sources of uncertainty in several ways. One way separates them to the aleatoric uncertainty and the epistemic uncertainly, the other divides them to predictive uncertainly, model & data uncertainly and the distributional uncertainly. The paper also discusses ways to estimate the certainly, including direct estimation, bayesian-based estimation and frequentist-based estimation, and "models with random parameters". The paper also introduces methods to handle the uncertainly in modeling including active learning, self-training, OOD detection, outlier detection and misclassification detection, as well as node level, edge level and graph level modeling and trustworthy GNN. The paper also mentions uncertainly induced by real-world application and future research directions, offering a survey.

**Audience:**

Yes

**Broader Impact Concerns:**

The publication of this paper is unlikely to introduce negative broader impacts.

**Claims And Evidence:**

Yes

**Requested Changes:**

1: The paper shall clarify the relationships between different categorization systems it uses in the paper. For example, which uncertainty belongs to aleatoric/epistemic uncertainty.
2: The paper shall modify the ambiguous definitions like "model with random parameters" to reduce potential misunderstanding.
3: The paper shall clarity the general definition of "uncertainty" and whether it includes graph attack/defense researches. Currently some methods like abnormalty detection are included but some others like robustness-enhanced models are not.

**Strengths And Weaknesses:**

The strength of the paper lies mainly in the following aspects.
1: It offers a thorough survey on many aspects of graph-uncertainty-related researches.
2: For each point listed in the survey, some references are offered so researchers could further investigate.
3: The paper also provides some potential future directions for graph-uncertainty-related researches.

The following points are the weaknesses of the paper.
1: It has ambiguous definition of "model with random parameters". I guess the authors want to mean "models with parameters being random variables", but the expression is likely to be misunderstood as parameter themselves being randomly generated, which shall not be the case. Any ambiguity is a large weakness for survey papers.
2: The paper uses two systems to classify different types of uncertainties, one system classifies by aleatoric uncertainty and epistemic uncertainty, where epistemic uncertainty includes model uncertainty and distributional uncertainty. Another system classifies by predictive uncertainty, model & data uncertainty and distributional uncertainty. The paper doesn't discuss the relationship between these two systems and it seems that the two systems introduced actually contradict with each other in definitions. For example, data uncertainty is put together with model uncertainty in system 2, but it shall act as part of the aleatoric uncertainty that belongs to different category as model uncertainty in system 1, causing ambiguity.
3: The paper mentions abnormality detection, but doesn't discuss any graph attack and defense methods. The paper shall clarify the meaning of uncertainty and whether it includes the graph pertuabtion attacks and robustness-enhancement defense methods.

---

> ### Author Response · Authors · 2024-06-17
>
> Thank you for your helpful suggestions! Here we present a point-wise rebuttal.
>
> **Response for Q1:** Thanks for the suggestion. This is indeed important. We have carefully gone thorugh the paper (e.g., Table 1) and modified this particular phrase.
>
> **Response for Q2:** We apologize for the ambiguity of classifications. We do not use two systems, but refer to them with different names following the norm in previous papers. In our paper, (total) uncertainty is equal to predictive uncertainty. Aleatoric uncertainty equals data uncertainty. Epistemic uncertainty includes model and distributional uncertainty.
>
> We have also clarified this particular issue in the caption of Figure 1.
>
> **Response for Q3:** We have included the literature on uncertainty for graph attack and robustness in Section 4.4. The general definition of "uncertainty" in this paper is given in the first paragraph of Section 2.
>
> We have made the requested changes. Please see the above answers and the revisions (in blue) in the revised version.

---

> > ### Comment · Reviewer_hKxZ · 2024-06-17
> > **Self-consistency on categorization of Uncertainty**
> >
> > Which category does model & data uncertainly belong to? Aleatoric or Epistemic? I suggest to separate the category(and use proper new definitions) if you find it belonging to both aleatoric uncertainty and epistemic uncertainty under the current definition.
> >
> > Self-consistency is vital in survey papers. You may be following the norm in previous papers, however that makes your paper not self-consistent. On that case you need to investigate more on referenced papers and decide which category in your paper they belong to according to your paper's definitions.

---

> > > ### Author Response · Authors · 2024-06-17
> > >
> > > Thank you for suggestion.
> > > In our paper, model uncertainty only belong to epistemic uncertainty, and data uncertainty is another name for aleatoric uncertainty.
> > >
> > > We are sorry for the confusion caused by referring them differently. The term "separation of aleatoric uncertainty and epistemic uncertainty" is commonly used and accepted in most papers. To avoid misundertanding, **we have changed all 'data uncertainty' to 'aleatoric uncertainty' (AU).**
> > >
> > > We need to further devide epistemic uncertainty (EU) into model uncertainty (MU) and distributional uncertainty (DU). EU = MU + DU. Futher, PU (Predictive Uncertainty) = AU + EU = AU + MU + DU. This relation is provided in Figure 2.
> > >
> > > In Section 3.4 Paragraph 4 "Model & Aleatoric uncertainty" and Paragraph 5 "Distributional Uncertainty", we introduce the corresponding evaluation metric for model, aleatoric (data) and distributional uncertainty.
> > >
> > > In many papers, they do not consider distributional uncertainty. So, according to those works, EU only contains and equals to MU. Some metrics can directly measure PU and MU, and use PU-MU to measure AU. In our proposed relation, PU - MU = AU + DU. If the predictions contains no distributional uncertainty, DU = 0, then there's no problem. But with the presence of DU (i.e., DU >0), the measure is not realiable. To avoid confusion, **we have revised to be consistent with epistemic uncertainty in tasks and applications (Section 4 & 5), as in most settings, distributional uncertainty is not apparent and set as none by default. Otherwise, if distributional uncertainty is clearly stated, e.g., as in OOD detection tasks, we stick with the model uncertainty. We also add this clarification in the last paragraph of Section 2, highlighted in red.**

---

### Review · Reviewer_65dG · 2024-04-24

**Summary Of Contributions:**

This paper presents a survey on uncertainty estimation, quantification and utilisation in graph machine learning.

**Audience:**

Yes

**Claims And Evidence:**

No

**Requested Changes:**

A major requested change is to introduce more of the Authors' own opinions in this survey. Try to make a clear stand about which methods should be used in which circumstances, with some levels of argumentation. Additionally, it would be useful to make a stand about whether certain papers are able to properly defend their hypothesis, or more investigation may be needed.

I think it might make sense to correct this attribution to DropEdge:

> Rong et al. (2019) introduce DropEdge, which randomly removes a specified number of edges from the graph by drawing independent Bernoulli random variables (with a constant rate) during each training iteration.

This is not completely factually correct. Rong et al. are the first to _theoretically analyse_ edge dropout, but they are definitely not the first paper to attempt edge dropout as a regulariser. I'm not sure what's the first paper to do it, but for example, even the GAT paper (Veličković et al., ICLR'18) leverages such an operator (without theoretical motivation, of course).

I also believe some sections would benefit from significantly more references being discussed.
The two sections that immediately come to mind as examples are **explainability** and **robustness**.

Presently, the explainability discussion in the paper doesn't go far beyond GNNExplainer. While GNNExplainer was the landmark paper in this space, there's been _a lot_ of developments since then. Papers such as GraphLIME, or GCExplainer, for example, are worth discussing.

Similarly, the robustness section appears to largely neglect work on robustness certification from the folks at TUM (e.g. Simon Geisler or Aleksandar Bojchevski, et al.) which would likely also be worthy of inclusion here.

**Strengths And Weaknesses:**

On the surface level, I think that this work delivers what it promises: there is a survey of a sizeable number of publications relating to GNNs and uncertainty, these publications are reasonably organised in a structure imposed by the authors, and some avenues for future work in the area are identified.

From that perspective, the effort invested by the authors could be considered worthy of a publication at TMLR.

However, I also consider TMLR's criteria to place surveys in a bit of a gray area.
I personally think the main utility from the surveys should come not only from an enumeration and organisation of a reference list, but also for conveying a useful opinion or argumentation that practitioners can follow when deciding which method to use.

Presently, the survey does not meet this criterion in my mind. In a lot of cases, a paper is simply introduced, briefly summarised in isolation, and then the survey simply moves on to the next one. It is very rare that I see the authors' own opinions and anecdotal experiences introduced, and as a result this wouldn't necessarily be a survey that would give a lot of value beyond an organisational skeleton.

Additionally, I think that there are several places where additional references and contexts could have been provided to make the story more accurate and robust (see next section for suggestions).

---

> ### Author Response · Authors · 2024-06-17
>
> Thank you for these insightful suggestions! They are useful and interesting to discuss in our survey paper. Our responses and actions are as follows.
>
> **Response for Q1:**
> *When to apply which methods*:
>
> In regards to how to apply those methods in tasks and applications, we believe this depends on the circumstances and especially on the followings: 1) the source of uncertainty, 2) example-wise or popoluation-level uncertainty, 3) computational efficiency, especially the difficulty of deploying them to existing prediction frameworks (ad-hoc or post-hoc), and 4) whether they are reliable and robust across diverse distributions. In the revised version, we have added a flowchart to recommend readers with appropriate methods in Figure 3, accompanied by detailed illustrations in Section 3.4. To make our opinons more clear, we have also revised and added more details about the strengths and weaknesses of each quantification methods in Section 3.1-3.3.
>
> *Investigation into hypothesis*:
>
> We have emphasized the assumptions in the paper particularly along the lines about the inconsistency between the real data distribution and the assumed distribution. We have further discussed Frequentist-based methods in Section 3.1, which has simple exchangeability assumption, which is crucial for the validity of these methods. Regarding other methods, due to lack of ground-truth for uncertainty and the complexity of using deep learning-based methods, we think it is difficult to provide rigorious proofs or justifications, and instead the assumtions are often supported with emperical evidence in the form of improved task performance. For example, in traffic-realted applications with GNNs, we have introduced Prob-GNN, which compares the prediction performance based on different assumptions on data distributions.
>
> Furthermore, different papers hypothesize on applicability of different types of uncertainty for each task, as evidenced by their quantification methods. This is also validated by the performance in downstream tasks. However, there is a gap in understanding whether performance enhancements are solely due to the selection of quantification methods or if they ideally result from an appropriate match between quantification methods and corresponding uncertainty types, where the uncertainty type itself aids the task. Bridging this gap requires ground-truth data or a unified evaluation metric, which is currently lacking.
>
> We have added this discussion in Section 6 Paragraph 3.
>
> **Response for Q2:** Thank you for mentioning this. We have revised it accordingly in Section 4.3 Paragraph 3:
>
> While Veličković et al. (2018) has applied a similar dropout technique on edge attentions, Rong et al. (2019) first formally presented the formulation of DropEdge. It randomly removes a specified number of edges from the graph by drawing independent Bernoulli random variables (with a constant rate) during each training iteration. Rong et al. (2019) theoretically and empirically show that DropEdge, as a data augmenter and a message-passing reducer, can alleviates the problems of over-fitting and over-smoothing.
>
> **Response for Q3:** We have revised section 4.4 and added more references in blue regarding explainability and robustness.
>
> **Response for Q4:** Please see the above answer. We have revised section 4.4 and added more references in blue regarding explainability.
>
> **Response for Q5:** Please see the answer for Q3. We have revised Section 4.4 and added more references in blue regarding robustness.

---

### Review · Reviewer_s3gv · 2024-06-10

**Summary Of Contributions:**

The paper offers a detailed examination of uncertainty in graph neural networks (GNNs), focusing on its identification, quantification, and utilization to improve the reliability and effectiveness of GNNs across various applications. It adeptly compares and synthesizes existing theories and methods related to graph uncertainty, integrating this uncertainty into graph learning tasks. Furthermore, the paper bridges various sub-communities within the GNN field, laying a foundation for future research avenues.

**Audience:**

Yes

**Claims And Evidence:**

No

**Requested Changes:**

See above

**Strengths And Weaknesses:**

**Strengths:**

1. **Comprehensive Coverage:** The survey extensively covers the topic of uncertainty in GNNs, ranging from the identification of uncertainty sources to their implementation in real-world scenarios. This broad scope makes it an indispensable resource for researchers in the field.
2. **Structured Framework and Organization:** The authors effectively structure a diverse range of literature into a coherent framework. This organization facilitates comprehension and allows readers to effortlessly navigate the complex aspects of GNN uncertainty.
3. **Insightful Analysis and Future Directions:** The paper not only consolidates existing knowledge but also identifies critical research gaps and suggests potential future research directions, such as fine-grained uncertainty in graphs and task-specific uncertainty considerations. These insights are invaluable for propelling the field forward and steering future studies.

**Areas for Improvement:**

1. **Complexity for Novices:** The paper may pose challenges for newcomers to the field of uncertainty in GNNs. Including more foundational background information could help make the survey more accessible to a wider audience.
2. **Need for Empirical Evidence:** While the survey is rich in discussions and general observations, incorporating empirical analyses or case studies could enhance its practical impact and provide concrete examples of the theories discussed.

---

> ### Author Response · Authors · 2024-06-17
>
> Thank you for your constructive suggestions!
>
> **Response for Q1:** This survey mainly focuses on uncertainty in GNNs. We have provided [1,2,3] in Section 1 as related works of uncertainty in deep learning, and the different sources of uncertainty in Section 2. For other related background, we encourage readers to investigate [4] for a detailed introduction to GNNs. We have added the background in Section 1 Paragraph 4.
>
> **Response for Q2:** Thank you for the suggestion! We have provided concrete applications and their corresponding related work in Section 5.
>
> For surveys, we have followed the guidelines of TMLR (https://jmlr.org/tmlr/editorial-policies.html). It says the folllowing: "surveys that draw new connections, highlight trends, and suggest new problems in an area".
>
> Nonetheless it will be an interesting future direction to add empirical analyses of different methods and prepare a benchmarking study.
>
> [1] Abdar, M., Pourpanah, F., Hussain, S., Rezazadegan, D., Liu, L., Ghavamzadeh, M., ... & Nahavandi, S. (2021). A review of uncertainty quantification in deep learning: Techniques, applications and challenges. Information fusion, 76, 243-297.
>
> [2] Hüllermeier, E., & Waegeman, W. (2021). Aleatoric and epistemic uncertainty in machine learning: An introduction to concepts and methods. Machine learning, 110(3), 457-506.
>
> [3] Gawlikowski, J., Tassi, C. R. N., Ali, M., Lee, J., Humt, M., Feng, J., ... & Zhu, X. X. (2023). A survey of uncertainty in deep neural networks. Artificial Intelligence Review, 56(Suppl 1), 1513-1589.
>
> [4] Wu, Z., Pan, S., Chen, F., Long, G., Zhang, C., & Philip, S. Y. (2020). A comprehensive survey on graph neural networks. IEEE transactions on neural networks and learning systems, 32(1), 4-24.

---

### Author Response · Authors · 2024-06-17
**Submission of Revised Paper and Rebuttal**

Thanks to all the reviewers for their helpful suggestions. We have incorporated the suggestions in our revised version. The revision is indicated in blue in the draft. We mainly provide more comments and opinions of existing methods, rephrase some ambiguous statements, and add more related works and backgrounds. We also present a point-wise rebuttal for each reviewer.

---

### Decision · Action_Editor_1J56 · 2024-09-29

**Recommendation:** Accept as is

**Comment:**

The updated manuscript already address most of the reviewers' comments.

**Audience:**

It is suitable for TMLR.

**Claims And Evidence:**

The paper offers a detailed examination of uncertainty in graph neural networks (GNNs), focusing on its identification, quantification, and utilization to improve the reliability and effectiveness of GNNs across various applications.

Two complaints about the paper are:
1. The organization of the work does not follow a clear tree structure. Instead there are overlapping.
2. There is too much listing of paper without Author's own opinion/insights summarizing the paper.

The overlapping organization would not be a major issue since the taxonomy may not be non-overlapping.
The second issue is less severe if authors could further revise a bit. The current form is also acceptable.

Overall, the paper could be accepted by TMLR as a survey.